# Maximizing the robust margin provably overfits on noiseless data

**Konstantin Donhauser** [* 1]  **Alexandru Țifrea** [* 1]  **Michael Aerni** [1]  **Reinhard Heckel** [2 3]  **Fanny Yang** [1]

## Abstract

Numerous recent works show that overparameterization implicitly reduces variance, suggesting vanishing benefits for explicit regularization in high dimensions. However, this narrative has been challenged by empirical observations indicating that adversarially trained deep neural networks suffer from robust overfitting. While existing explanations attribute this phenomenon to noise or problematic samples in the training data set, we prove that even on entirely noiseless data, achieving a vanishing adversarial logistic training loss is suboptimal compared to regularized counterparts.

## 1. Introduction

A modern narrative in machine learning suggests that regularization is superfluous for good performance of large overparameterized models. This perspective is motivated by theoretical and experimental findings that analyze the population risk in a setup where the training and test data are drawn from the same distribution (see Hastie et al. (2019); Bartlett et al. (2020); Yang et al. (2020); Nakkiran et al. (2020) and references therein). However, the successful adoption of machine learning models in real-world applications crucially hinges on the models' robustness to adversarial attacks or distribution shifts. A popular method to achieve low *robust risk* is to minimize a robust training loss, for example using adversarial training (Goodfellow et al., 2015). Mounting empirical evidence suggests that when minimizing the robust loss, the narrative that regularization is superfluous is incorrect: additional regularization such as early-stopped adversarial training often leads to more robust generalization (Rice et al., 2020; Sagawa et al., 2020a;b). For example, Rice et al. (2020) show that overparameterized deep neural networks that are adversarially trained with $\ell_\infty$ perturbations benefit from early stopping on image data

---
[*]Equal contribution  [1]ETH Zurich  [2]Rice University  [3]Technical University of Munich.  Correspondence to: Konstantin Donhauser <konstantin.donhauser@ethz.ch>, Alexandru Țifrea <tifreaa@inf.ethz.ch>.

*Accepted by the ICML 2021 workshop on A Blessing in Disguise: The Prospects and Perils of Adversarial Machine Learning.* Copyright 2021 by the author(s).

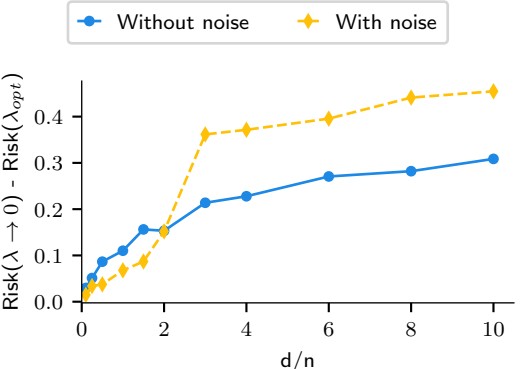

*Figure 1.* The robust risks for the robust max-margin solution ($\lambda \to 0$) are strictly higher than for a regularized minimizer ($\lambda_{opt} > 0$) even in the overparameterized regime $d \gg n$ and for noiseless training data (see detailed setting in Section 2 and experimental details in Appendix C). This illustrates how adversarial training benefits from regularization.

sets. This phenomenon is sometimes referred to as *robust overfitting*.

For noisy settings, previous works have proposed the following explanation for robust overfitting: since the robust risk amplifies estimation errors, its variance is larger, and regularization can hence be beneficial for robust generalization (Sanyal et al., 2021). However, Figure 4 in Appendix B shows that even when estimating mostly noiseless signals, robust overfitting persists! This observation does not only apply to image data but also to simple linear models: Figure 1 shows the gap in performance between $\ell_\infty$-robust logistic regression regularized by a ridge penalty with optimal parameter $\lambda_{opt}$ and the robust max-margin solution.

Existing theoretical results neither predict nor explain those observations. In particular, it is perhaps unexpected that for noiseless data, strictly sacrificing robust data fit by increasing the weight of the ridge penalty term leads to better generalization. In this paper, we show for robust logistic regression and a sparse ground truth that in the high-dimensional regime, where $d > n$, maximizing the robust margin leads to suboptimal robust generalization. Specifically,

- We prove that a strictly positive ridge parameter leads to a systematic improvement in generalization compared to the robust max-margin solution. Our results provide the first rigorous proof for robust overfitting even in the absence of noise.

- We experimentally show a perhaps unexpected effect of noise in the training procedure: noisy data might not be robustly linearly separable, rendering the robust max-margin estimator infeasible and hence inducing spurious regularization effects.

## 2. Risk minimization framework

In this section, we describe the setup for our theoretical analysis of adversarial training with the logistic loss. We define the data generating process, the standard and robust population risks, and formally introduce the estimators that we analyze.

### 2.1. Problem setting

We consider a discriminative data-generating model for classification with covariates (or features) $x \in \mathbb{R}^d$ drawn from an isotropic Gaussian distribution $x \sim \mathcal{N}(0, I_d)$. In this paper we focus primarily on training with noiseless data, namely we observe deterministic labels given by $y = \mathrm{sgn}\langle\theta^\star, x\rangle \in \{-1, +1\}$ for all covariates $x \in \mathbb{R}^d$, where the fixed vector $\theta^\star$ denotes the ground truth. We consider the scenario where the ground truth has inherent low-dimensional structure in the form of sparsity and pick $\theta^\star = (1, 0, \ldots, 0)^\top$. As we discuss later in this section, $\ell_\infty$ robustness and sparsity via its convex surrogate, the $\ell_1$ norm, are related objectives.

The results in this paper are of asymptotic nature and hold when $d/n \to \gamma$ as both the dimensionality $d$ and the number of samples $n$ tend to infinity. This high-dimensional regime is widely studied in the literature (Bühlmann & Van De Geer, 2011; Wainwright, 2019) as it yields precise predictions for many real-world problems where both the input dimension and the data set size are large. It is also the predominant setting considered in previous theoretical papers that discuss overparameterized linear models (Dobriban & Wager, 2018; Hastie et al., 2019; Ali et al., 2020; Deng et al., 2021; Javanmard et al., 2020; Javanmard & Soltanolkotabi, 2020; Sur & Candès, 2019).

### 2.2. Robust risk for evaluation

The broad application of ML models in real-world decision-making processes increases requirements on their robustness. For example, for the image domain, robust classifiers should yield the same prediction when an image is attacked via an additive imperceptible $\ell_\infty$-perturbation that does not change the ground truth label. In this case, the estimator which has zero *standard* population risk also achieves zero *robust* population risk. Transferred to linear classification, we require such additive *consistent* perturbations to be orthogonal to $\theta^\star$, that is $\delta \in \mathcal{U}_c(\epsilon) := \{\delta \in \mathbb{R}^d : \|\delta\|_\infty \leq \epsilon \text{ and } \langle\theta^\star, \delta\rangle = 0\}$. We hence evaluate the adversarially robust risk of a parameter $\theta$ with respect to consistent $\ell_\infty$-

perturbations, defined as follows:

$$\mathbf{R}_\epsilon(\theta) := \mathbb{E}_{X \sim \mathbb{P}} \min_{\delta \in \mathcal{U}_c(\epsilon)} \mathbb{1}_{\mathrm{sgn}(\langle\theta, X+\delta\rangle) \neq \mathrm{sgn}(\langle\theta^\star, X\rangle)}, \quad (1)$$

where the expectation is taken over the marginal feature distribution $\mathbb{P}$ and $\mathbb{1}$ is the indicator function.

For $\epsilon = 0$ we obtain the population 0-1 risk, also called standard risk, denoted by $\mathbf{R}(\theta)$. For the 1-sparse ground truth $\theta^\star$, the risks have a closed-form expression given by Lemma A.1 in Appendix A.1.

### 2.3. Interpolating and regularized estimator

We say that the data is *robustly separable* when the robust max-margin solution exists. Its direction is given by

$$\hat{\theta}_0 := \arg\min_\theta \|\theta\|_2 \text{ such that for all } i, \quad (2)$$

$$\max_{\delta \in \mathcal{U}(\epsilon)} y_i\langle\theta, x_i + \delta\rangle \geq 1.$$

Robust separability is guaranteed for noiseless data and for consistent perturbations $\mathcal{U}(\epsilon) = \mathcal{U}_c(\epsilon)$ that are orthogonal to the ground truth $\theta^\star$. Unfortunately, minimization with respect to consistent perturbations in Equation (2) requires full knowledge of the ground truth during training, thus leaking information about $\theta^\star$. Instead, many papers to date (e.g. Javanmard et al. (2020); Javanmard & Soltanolkotabi (2020)) consider the unrestricted (and hence inconsistent) $\ell_\infty$-perturbation set $\mathcal{U}_{ic}(\epsilon) := \{\delta \in \mathbb{R}^d : \|\delta\|_\infty \leq \epsilon\}$. However, when training with inconsistent perturbations, it is possible that the perturbed data crosses the true decision boundary. Thus, a minimizing solution might fit wrong labels even if the actual training samples are noiseless.

We want to study the purely noiseless setting and hence focus primarily on adversarial training with consistent perturbations. According to classical wisdom, this setting is also the one in which we expect regularization to help the least. We revisit inconsistent adversarial training in Section 4.2 and show that its effects can at least spuriously improve the robust risk.

We compare the robust max-margin interpolator with the ridge-regularized robust logistic regression. We can write the minimizer of the regularized $\ell_\infty$ robust logistic loss as follows:

$$\hat{\theta}_\lambda := \arg\min_\theta \frac{1}{n} \sum_{i=1}^n \max_{\delta \in \mathcal{U}(\epsilon)} \log(1 + e^{-\langle\theta, x_i+\delta\rangle y_i}) + \lambda\|\theta\|_2^2.$$
$$(3)$$

Notice that for robustly separable data, the direction of the robust max-margin solution aligns with that of the minimizer $\hat{\theta}_\lambda$ for $\lambda \to 0$, i.e., $\hat{\theta}_0 = \lim_{\lambda \to 0} \hat{\theta}_\lambda$. This fact follows from the results in (Rosset et al., 2004) when using the closed form expression for the consistent robust risk (1). Since the robust risk is independent of the estimator norm, we refer

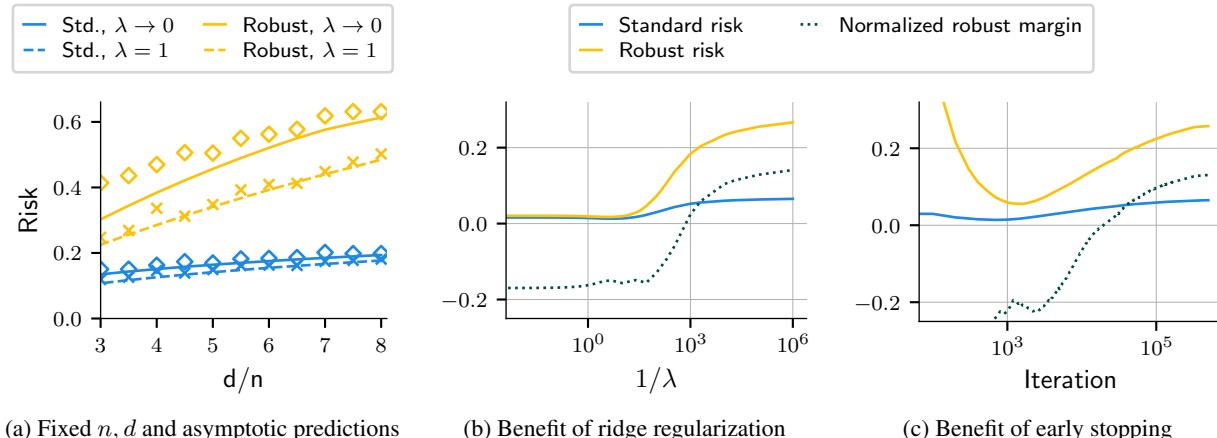

*Figure 2.* (a) Robust and standard risks of the robust max-margin ($\lambda \to 0$) and robust regularized minimizer ($\lambda = 1$) using $\epsilon = 0.05$ as a function of the overparameterization ratio $d/n$ for simulations (markers) and asymptotic theoretical predictions from Theorem 3.1 (lines). (b),(c) Normalized robust margins and risks of empirical simulations using $\epsilon = 0.1$ and $d/n = 8$, with respect to: (b) increasing $1/\lambda$; and (c) gradient descent iterations when minimizing Equation (3) using $\lambda = 0$. The normalized robust margin is defined as $\min_i \min_{\delta \in \mathcal{U}_c(\epsilon)} y_i \langle \theta, x_i + \delta \rangle / \|\theta\|_2$. Both ridge regularization and early stopping yield superior robust and standard risks. All experiments use $n = 1000$ and consistent $\ell_\infty$ perturbations. See Appendix C for experimental details.

to the robust max-margin solution as the normalized vector $\hat{\theta}_0 / \|\hat{\theta}_0\|_2$ and for brevity simply call it the max-margin solution.

Finally, we remark that for standard logistic regression, gradient descent converges to the same direction as the max-margin solution (Ji & Telgarsky, 2019; Soudry et al., 2018), and, as argued in (Javanmard & Soltanolkotabi, 2020), this result can also be extended to adversarial training (3).

## 3. Regularization for robustly separable data

In this section, we prove that the robust max-margin solution (2) overfits since it generalizes worse than the regularized estimator (3) with optimally chosen $\lambda > 0$ for entirely noiseless training data. We derive precise asymptotic predictions that show a systematic benefit of ridge regularization for robust logistic regression as $d, n \to \infty$ and $d/n \to \gamma$. We further show that early-stopped gradient descent on the unregularized robust logistic loss in Equation (3) yields much smaller population risks than the robust max-margin solution at convergence.

### 3.1. Benefits of ridge regularization

We now present an informal statement describing the asymptotic standard and robust risk for linear classification with the logistic loss as $d, n \to \infty$. We refer to Appendix D for the precise statement and the proof, which is inspired by the works of Javanmard & Soltanolkotabi (2020); Salehi et al. (2019) and uses the Convex Gaussian Minimax Theorem (CGMT) (Gordon, 1988; Thrampoulidis et al., 2015).

**Theorem 3.1** (Informal)**.** *Assume that $\epsilon = \epsilon_0/\sqrt{d}$ for some constant $\epsilon_0$ independent of $n, d$ and $\theta^\star = (1, 0, \cdots, 0)$.*

*Then, the robust and standard risks of the estimator $\hat{\theta}_\lambda$ from Equation (2),(3) with respect to consistent attacks and $\lambda \geq 0$ converge in probability as $d, n \to \infty$, $d/n \to \gamma$*

$$\boldsymbol{R}(\hat{\theta}_\lambda) \to \frac{1}{\pi} \arccos\left(\frac{\nu_\parallel^\star}{\nu^\star}\right)$$

$$\boldsymbol{R}_\epsilon(\hat{\theta}_\lambda) \to \boldsymbol{R}(\hat{\theta}_\lambda) + \frac{1}{2}\mathrm{erf}\left(\frac{\epsilon_0 \delta^\star}{\sqrt{2}\nu^\star}\right) + I\left(\frac{\epsilon_0 \delta^\star}{\nu^\star}, \frac{\nu_\parallel^\star}{\nu^\star}\right)$$

*with* $\mathrm{erf}$ *the error function,*

$$I(t, u) := \int_0^t \frac{1}{\sqrt{2\pi}} \exp\left(-\frac{x^2}{2}\right) \mathrm{erf}\left(\frac{xu}{\sqrt{2(1-u^2)}}\right) dx$$

*and where* $\nu^\star = \sqrt{(\nu_\perp^\star)^2 + (\nu_\parallel^\star)^2}$, *and* $\nu_\perp^\star, \nu_\parallel^\star, \delta^\star$ *are the unique solution of a scalar optimization problem specified in Appendix D that depends on* $\theta^\star, \gamma, \epsilon_0$ *and* $\lambda$.

We illustrate the theorem's claim in Figure 2a where we show the standard and robust risks obtained by the asymptotic theoretical predictions for $d, n \to \infty$ and the risks obtained from simulations for $n = 1000$ and $\epsilon = 0.1$. The theoretical curves are obtained by solving the scalar optimization problem specified in Appendix D; the empirical setup is described in Appendix C. We observe that the precise asymptotics indeed predict benefits of regularization for robust logistic regression on noiseless training data. This trend is further supported by our simulation results closely following the theoretical curves. Furthermore, in Figure 2b, we observe for a fixed overparameterization ratio $d/n = 8$ how optimizing the robust logistic loss well beyond $100\%$ robust training accuracy (i.e. the robust margin becomes positive) substantially hurts generalization.

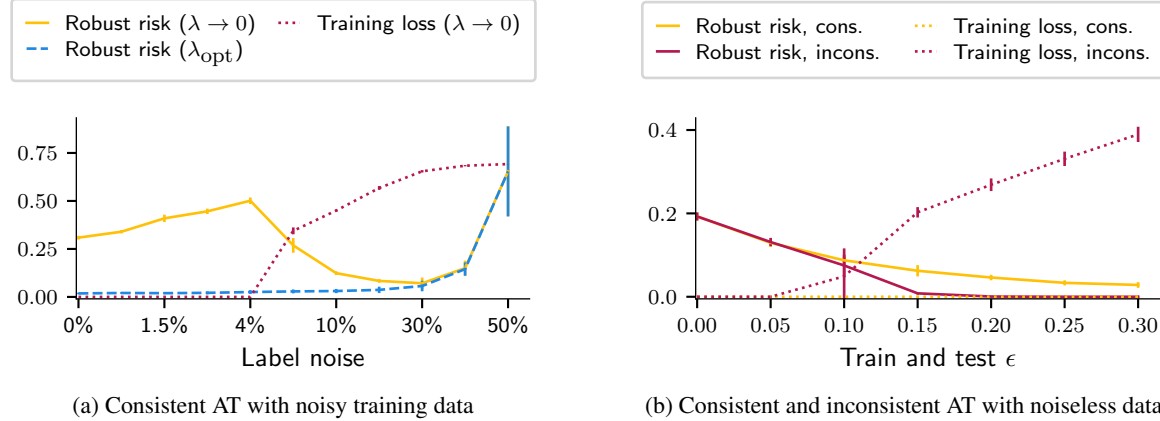

(a) Consistent AT with noisy training data

(b) Consistent and inconsistent AT with noiseless data

*Figure 3.* (a) Training loss and robust risks with respect to increasing training label noise for $\epsilon = 0.1$, $d = 8000$ and $n = 1000$. We observe for unregularized estimators ($\lambda \to 0$) that, counterintuitively, moderate amounts of label noise decrease the robust risk by avoiding the robust max-margin solution. While this might spuriously imply that injecting label noise increases robustness, estimators with optimal ridge parameter ($\lambda_{\mathrm{opt}}$) still outperform their unregularized counterparts in terms of robust risk. (b) Training loss and risks with respect to increasing train and test $\epsilon$ for $d = 500$, $n = 1000$ and $\lambda \to 0$. In contrast to training with consistent attacks, unregularized inconsistent adversarial training (AT) does not achieve vanishing training loss for large enough $\epsilon$, and hence leads to a smaller robust risk.

### 3.2. Benefits of early stopping

In addition to varying the ridge coefficient $\lambda$, we observe the same trends as for $1/\lambda \to \infty$ on the optimization path when training with gradient descent as the iterations $t \to \infty$. Figure 2c indicates that early stopped gradient descent can improve the robust risk for logistic regression, similar to ridge regularization. Both early stopping and ridge regularization avoid the max-margin solution $t \to \infty$ and $\lambda \to \infty$ respectively, and yield an estimator with significantly lower standard and robust risk.

## 4. The benefits of avoiding robust separability

In the previous section we focused on noiseless settings and studied the generalization performance of regularized estimators that do not maximize the robust margin. In this section, we show how, surprisingly, as a consequence of avoiding robust separability, adding noisy labels in the training loss also leads to an estimator with better robust generalization than the max-margin solution of the corresponding noiseless problem. However, we note that explicit regularization is still the preferred choice of avoiding the max-margin solution as it yields a significantly lower robust risk.

### 4.1. Avoiding the max-margin solution via label noise

We first consider the case where we introduce label noise to the training data. Figure 3a shows the robust and standard risk together with the training loss of estimator $\hat{\theta}_\lambda$ (3) for $\lambda \to 0$ with varying percentages of flipped labels. For low noise levels, the data is robustly separable and the training loss vanishes, yielding the max-margin solution (2). For high enough noise levels, the constraints in (2) become infeasible and the training loss of the resulting estimator starts

to increase. This effectively induces implicit regularization as the robust risk at this point starts to decrease again. While it is well known that covariate noise can act as an implicit regularization (Bishop, 1995), in contrast to common intuition, we show that the performance can also improve when introducing wrong labels in the training loss as it prevents the estimator from converging to the max-margin solution as $\lambda \to 0$. While this spuriously implies that label noise benefits robustness, the estimator with optimal ridge parameter $\lambda_{\mathrm{opt}}$ always outperforms the unregularized estimator, even if the data is not robustly-separable.

### 4.2. Avoiding the max-margin solution via inconsistent training

For large $\epsilon$ and inconsistent perturbations $\mathcal{U}_{ic}(\epsilon)$, even noiseless training data might not be robustly separable, rendering the max-margin solution infeasible for $\lambda \to 0$. Consequently, this leads to a similar effect as adding label noise as discussed in Section 4.1. Indeed, Figure 3b reveals that the resulting estimator trained with inconsistent adversarial perturbations achieves a better performance than the corresponding max-margin estimator (2).

## 5. Conclusions

In this paper, we prove that on high-dimensional input data, the ridge-regularized robust logistic loss yields a solution that is more robust than the unregularized robust max-margin estimator, even on entirely noiseless data. We further show experimentally that early-stopped gradient descent yields similar benefits and discuss other phenomena that unexpectedly improve robust generalization by avoiding the max-margin solution.

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

# A. Setting

In Section A.1 we derive closed-form expressions of the standard and robust risks (1). Furthermore, Section A.2 discusses that the robust risk (1) upper-bounds the worst case risk under distributional mean shifts.

## A.1. Closed-form of robust risk for classification

Similarly to linear regression, we can express the robust and standard risk for the linear classification model in Section 3 as stated in the following lemma.

**Lemma A.1.** *Assume that $\mathbb{P}_X$ is the isotropic Gaussian distribution and $\theta^\star = (1, 0, \cdots, 0)^\top$. Then,*

1. *For any non-decreasing loss $\ell : \mathbb{R} \to \mathbb{R}$ we have*

$$\max_{\delta_i \in \mathcal{U}_\infty(\epsilon)} \ell(y_i \langle x_i + \delta_i, \theta \rangle) = \ell(y_i \langle x_i, \theta \rangle - \epsilon \|\Pi_\perp \theta\|_1). \tag{4}$$

2. *For the* 0-1 *loss the robust risk* (1) *with respect to $\ell_\infty$ perturbations is given by*

$$\mathbf{R}_\epsilon(\theta) = \frac{1}{\pi} \arccos\left( \frac{\langle \theta^\star, \theta \rangle}{\|\theta\|_2} \right) + \frac{1}{2} \mathrm{erf}\left( \frac{\epsilon \|\Pi_\perp \theta\|_1}{\sqrt{2}\|\theta\|_2} \right) + I\left( \frac{\epsilon\|\Pi_\perp \theta\|_1}{\|\theta\|_2}, \frac{\langle \theta^\star, \theta \rangle}{\|\theta\|_2} \right), \tag{5}$$

*with*

$$I(t, u) := \int_0^t \frac{1}{\sqrt{2\pi}} \exp\left( -\frac{x^2}{2} \right) \Phi\left( \frac{xu}{\sqrt{2}\sqrt{1 - u^2}} \right) dx. \tag{6}$$

*Proof.* We first prove Eq (4). Because $\ell$ is non-increasing, we have

$$\max_{\delta_i \in \mathcal{U}_\infty(\epsilon)} \ell(y_i \langle x_i + \delta_i, \theta \rangle) = \ell(\min_{\delta_i \in \mathcal{U}_\infty(\epsilon)} y_i \langle x_i + \delta_i, \theta \rangle)$$

$$= \ell(y_i \langle x_i, \theta \rangle + \min_{\|\delta_i\|_\infty \leq \epsilon, \delta_i \perp \theta^\star} \langle \delta_i, \theta \rangle).$$

While minimization over $\delta$ has no closed-form solution in general, we note that for $\theta^\star = (1, 0, \cdots, 0)$, we get $\min_{\|\delta_i\|_\infty \leq \epsilon, \delta_i \perp \theta^\star} \langle \delta_i, \theta \rangle = -\epsilon \|\Pi_\perp \theta\|_1$ and Equation (4).

Let $\mathbb{1}\{E\}$ be the indicator function, 1 if the event $E$ holds. Since $\ell(\cdot) = \mathbb{1}_{\cdot \leq 0}$ is non-increasing we can use (4) and write

$$\mathbf{R}_\epsilon(\theta) = \mathbb{E}_X \max_{\delta \in \mathcal{U}_\infty(\epsilon)} \mathbb{1}\{\mathrm{sgn}(\langle X, \theta^\star \rangle)\langle X + \delta, \theta \rangle \leq 0\}$$

$$= \mathbb{E}_X \mathbb{1}\{\mathrm{sgn}(\langle X, \theta^\star \rangle)\langle X, \theta \rangle - \epsilon \|\Pi_\perp \theta\|_1 \leq 0\}.$$

Let $\widehat{\Pi}_\| := \frac{1}{\|\theta\|_2^2} \theta \theta^\top$ be the projection onto the subspace spanned by $\theta$ and $\widehat{\Pi}_\perp := I_d - \widehat{\Pi}_\|$ the projection onto the orthogonal subspace. Since $X$ is a vector with i.i.d. standard normal distributed entries, we equivalently have

$$\mathbf{R}_\epsilon(\theta) = \mathbb{E}_{Z_1, Z_2} \mathbb{1}\{Z_1 \mathrm{sgn}\left( Z_1 \|\widehat{\Pi}_\| \theta^\star\|_2 + Z_2 \|\widehat{\Pi}_\perp \theta^\star\|_2 \right) - \epsilon \frac{\|\Pi_\perp \theta\|_1}{\|\theta\|_2} \leq 0\}, \tag{7}$$

with $Z_1, Z_2$ two independent standard normal random variables. For brevity of notation, define $\nu = \epsilon \frac{\|\Pi_\perp \theta\|_1}{\|\theta\|_2}$ and $b(Z_1, Z_2) = \mathrm{sgn}\left( Z_1 \|\widehat{\Pi}_\| \theta^\star\|_2 + Z_2 \|\widehat{\Pi}_\perp \theta^\star\|_2 \right) =: \mathrm{sgn}(\beta^\top Z)$ with $\beta^\top = (\|\widehat{\Pi}_\| \theta^\star\|_2, \|\widehat{\Pi}_\perp \theta^\star\|_2)$.

Define the event $A = \{\mathrm{sgn}\left( Z_1 \|\widehat{\Pi}_\| \theta^\star\|_2 + Z_2 \|\widehat{\Pi}_\perp \theta^\star\|_2 \right) - \epsilon \frac{\|\Pi_\perp \theta\|_1}{\|\theta\|_2} \leq 0\}$. Because $Z_2$ is symmetric, the distribution of $Z_1 b(Z_1, Z_2)$ is symmetric, hence we can rewrite the risk

$$\mathbf{R}_\epsilon(\theta) = \underbrace{\mathbb{P}(b(Z_1, Z_2) \leq 0 | Z_1 \geq 0)}_{T_1} + \underbrace{\mathbb{P}(Z_1 \leq \nu, b(Z_1, Z_2) \geq 0 | Z_1 \geq 0)}_{T_2} \tag{8}$$

and derive expression for $T_1, T_2$ separately.

**Step 1: Proof for $T_1$**    Note that due to $\|\theta^\star\|_2 = 1$ we have $\|\beta\|_2 = 1$ and recall that $T_1 = \mathbb{P}(\beta^\top Z \leq 0 | Z_1 \geq 0)$. Using the fact that both $Z_1$ and $Z_2$ are independent standard normal distributed random variables, a simple geometric argument then yields that $T_1 = \frac{\alpha}{\pi}$ with $\alpha = \arccos\left(\frac{\beta_1}{\sqrt{\beta_1^2 + \beta_2^2}}\right) = \arccos(\beta_1)$. Noting that $\beta_1 = \|\widehat{\Pi}_\| \theta^\star\|_2 = \frac{\langle \theta^\star, \theta\rangle}{\|\theta\|_2}$ then yields $T_1 = \frac{1}{\pi}\arccos\left(\frac{\langle \theta^\star, \theta\rangle}{\|\theta\|_2}\right)$.

**Step 2: Proof for $T_2$**    First, assume that $\langle \theta^\star, \theta \rangle \geq 0$. We separate the event $\mathcal{V} = \{Z_1 \leq \nu, b(Z_1, Z_2) \geq 0\}$ into two events $\mathcal{V} = \mathcal{V}_1 \cup \mathcal{V}_2$

$$\mathcal{V}_1 = \{Z_1 \leq \nu, Z_2 \geq 0\} \ \text{ and } \ \mathcal{V}_2 = \{Z_1 \leq \nu, b(Z_1, Z_2) \geq 0, Z_2 \leq 0\}.$$

The conditional probability of the first event is directly given

$$\mathbb{P}(\mathcal{V}_1 | Z_1 \geq 0) = \mathbb{P}(Z_2 \geq 0)\mathbb{P}(Z_1 \leq \nu | Z_1 \geq 0) = \frac{1}{2}\mathrm{erf}\left(\frac{\nu}{\sqrt{2}}\right).$$

Hence it only remains to find an expression for $\mathbb{P}(\mathcal{V}_2 | Z_1 \geq 0)$. Letting $\mu$ denote the standard normal distribution, we can write

$$\mathbb{P}(Z_1 \leq \nu, Z_2 \leq 0, b(Z_1, Z_2) \geq 0 | Z_1 \geq 0) = 2\int_0^\nu \int_0^{\frac{\beta_1 x}{\beta_2}} d\mu(y)d\mu(x) = \int_0^\nu \frac{1}{2}\mathrm{erf}\left(\frac{\beta_1 x}{\beta_2}\right)d\mu(x).$$

Together with Step 1, Equation (5) follows by noting that $\beta_1^2 + \beta_2^2 = 1$. Finally, the proof for the case where $\langle \theta^\star, \theta \rangle \leq 0$ follows exactly from the same argument and the proof is complete. $\qquad\square$

### A.2. Distribution shift robustness and consistent adversarial robustness

In this section we rigorously introduce distribution shift robustness and show the relation to consistent $\ell_p$ adversarial robustness for certain types of distribution shifts.

When learned models are deployed in the wild, the i.i.d. assumption does not always hold. That is, the test loss might be evaluated on samples from a slightly different distribution. Shifts in the mean of the covariate distribution is a standard intervention studied in the invariant causal prediction literature (Bühlmann et al., 2020; Chen & Bühlmann, 2020). For mean shifts in the null space of the ground truth $\theta^\star$ we define an alternative evaluation metric that we refer to as the *distributionally robust risk* defined as follows:

$$\tilde{\mathbf{R}}_\epsilon(\theta) := \max_{\mathbb{Q} \in \mathcal{V}_q(\epsilon; \mathbb{P})} \mathbb{E}_{X \sim \mathbb{Q}}\ell_{\text{test}}(\langle \theta, X + \delta\rangle, \langle \theta^\star, X\rangle), \ \text{ with}$$

$$\mathcal{V}_p(\epsilon; \mathbb{P}) := \{\mathbb{Q} \in \mathcal{P} : \|\mu_\mathbb{P} - \mu_\mathbb{Q}\|_p \leq \epsilon \text{ and } \langle \mu_\mathbb{P} - \mu_\mathbb{Q}, \theta^\star\rangle = 0\}$$

where $\mathcal{V}_p$ is the neighborhood of mean shifted probability distributions.

A duality between distribution shift robustness and adversarial robustness has been established in earlier work such as (Sinha et al., 2018) for general convex, continuous losses $\ell_{\text{test}}$. For our setting, the following lemma holds.

**Lemma A.2.** *For any $\epsilon \geq 0$ and $\theta$, we have $\tilde{\mathbf{R}}_\epsilon(\theta) \leq \mathbf{R}_\epsilon(\theta)$.*

*Proof.* The proof follows directly from the definition and consistency of the perturbations $\mathcal{U}_p(\epsilon)$ and orthogonality of the mean shifts for the neighborhood $\mathcal{V}_p$. By defining the random variable $w = x - \mu_\mathbb{P}$ for $x \sim \mathbb{P}$ we have the distributional equivalence

$$x' = \mu_\mathbb{P} + \delta + w \stackrel{d}{=} x + \delta$$

for $x' \sim \mathbb{Q}$ and $x \sim \mathbb{P}$ with $\mu_\mathbb{Q} - \mu_\mathbb{P} = \delta$ and hence

$$\tilde{\mathbf{R}}_\epsilon(\theta) = \max_{\mathbb{Q} \in \mathcal{V}_p(\epsilon)} \mathbb{E}_{x \sim \mathbb{Q}}\ell_{\text{test}}(\langle \theta, x\rangle, \langle \theta^\star, x\rangle) = \max_{\|\delta\|_p \leq \epsilon, \delta \perp \theta^\star} \mathbb{E}_{x \sim \mathbb{P}}\ell_{\text{test}}(\langle \theta, x + \delta\rangle, \langle \theta^\star, x\rangle)$$

$$\leq \mathbb{E}_{x \sim \mathbb{P}}\max_{\|\delta\|_p \leq \epsilon, \delta \perp \theta^\star}\ell_{\text{test}}(\langle \theta, x + \delta\rangle, \langle \theta^\star, x\rangle) = \mathbf{R}_\epsilon(\theta)$$

where the first line follows from orthogonality of the mean-shift to $\theta^\star$. $\qquad\square$

## B. Neural networks on sanitized binary MNIST

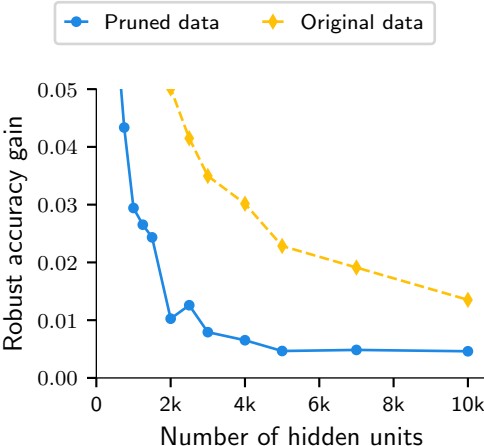

*Figure 4.* Increase in robust test accuracy when using early stopping for single hidden layer neural networks with different widths on a subset of MNIST classes 1 vs 3. The curves depict the differences of the test robust accuracy between the best and last iteration of training. Even if we discard training samples that are difficult to fit (pruned data), early stopping yields strictly positive benefits for the robust accuracy.

Figure 4 shows that robust overfitting in the overparameterized regime also occurs for single hidden layer neural networks on an image classification problem that we chose to be arguably devoid of noise. We consider binary classification on MNIST classes 1 vs 3 and further reduce variance by removing "difficult" samples: We train networks of width $p \in \{10, 100, 10k\}$ on the full MNIST training dataset and discard all images that take at least one of the models more than 100 epochs of training to learn. While some recent work argues that such sanitation procedures can effectively mitigate robust overfitting (Dong et al., 2021), we still observe a significant increase from the final test robust accuracies to the best (early stopped) ones in Figure 4.

We train all networks on a subset of $n = 2000$ training samples using vanilla mini-batch stochastic gradient descent with learning rate $\nu_p = \sqrt{0.1/p}$ that we multiply by 0.1 after 300 epochs. This learning rate schedule minimizes the training loss efficiently; we did not perform tuning using test or validation data. For the robust test error, we approximate worst-case $\ell_\infty$ perturbations using 10-step SGD attacks on each test sample.

## C. Experimental details

In this section we provide additional details on our experiments. All our code including instructions and hyperparameters can be found here: `https://github.com/michaelaerni/interpolation_robustness`.

### C.1. Empirical predictions

If not mentioned otherwise, all empirical simulations use noiseless i.i.d. samples from our synthetic data model as described in Section 2.1. Whenever we simulate label noise, we flip a fixed percentage of all training labels chosen uniformly at random. However, we calculate all risks in closed-form without noise and, in the robust case, with consistent perturbations. We approximate the integral for the robust 0-1 risk in Theorem 3.1 using a numerical integral solver since we cannot obtain a solution analytically.

We fit all logistic regression models except in Figure 2c by minimizing the (regularized) robust logistic loss (3) using the *CVXPY* library in combination with the *Mosek* convex programming solver. To obtain the max-margin solution whenever feasible for $\lambda \to 0$, we optimize the constrained problem (2) directly since (3) with $\lambda = 0$ has many optimal solutions.

In Figure 1, we use $\epsilon = 0.1$ for the robust risk and consistent adversarial training on $n = 1000$ samples. In the noisy case, we flip 2% of all training labels. For Figure 2c, we run zero-initialized gradient descent on the unregularized loss ($\lambda = 0$) for $500k$ iterations. We start with a small initial step size of $0.01$ that we double every $30k$ steps until iteration $300k$.

## C.2. Theoretical predictions

In order to obtain the asymptotic theoretical predictions for logistic regression in Figure 2a corresponding to the empirical simulations with $n = 1000$ and $\epsilon = 0.05$, we obtain the solution of the optimization problems in Theorem D.1,D.2 with $\epsilon_0 = 0.05\sqrt{1000\gamma}$ by solving the system of equations in Corollary D.4,D.5 using *MATLAB*'s optimization toolbox where we approximate expectations via numerical integration.

## D. Details on Theorem 3.1

In this section we give a formal statement for Theorem 3.1. The results are based on the Convex Gaussian Minimax Theorem (CGMT)(Gordon, 1988; Thrampoulidis et al., 2015). The results presented in this section have similarities with the ones in (Javanmard & Soltanolkotabi, 2020), however, we study the discriminative data model with features drawn from a single Gaussian and 1-sparse ground truth. In contrast, in the paper (Javanmard & Soltanolkotabi, 2020), the authors study a generative data model with features drawn from two Gaussians. Furthermore, several papers study logistic regression for isotropic Gaussian features in high dimensions (Salehi et al., 2019; Sur & Candès, 2019), but the analysis is focused on the standard risk and the authors do not consider adversarial robustness.

An immediate consequence of the proof of Lemma A.1 is that the adversarial loss from Equation (3) for the 1-sparse ground truth writes as

$$\mathcal{L}_{\epsilon,\lambda}(\theta) = \frac{1}{n}\sum_{i=1}^{n}\ell_{\text{train}}(y_i\langle\theta, x_i\rangle - \epsilon\|\Pi_\perp\hat{\theta}\|_1) + \lambda\|\theta\|_2^2. \tag{9}$$

Let $\mathcal{M}_f(x, t) = \min_y \frac{1}{2t}(x - y)^2 + f(y)$ be the Moreau envelope and let $Z_\|, Z_\perp$ be two independent standard normal distributed random variables. We can now state Theorem D.1 which describes the asymptotic risk of $\hat{\theta}_\lambda$, for $\lambda > 0$, and for the asymptotic regime where $d, n \to \infty$. The proof of the theorem can be found in Appendix D.3.

**Theorem D.1.** *Assume that the we have i.i.d. random features $x_i$ drawn from an isotropic Gaussian with noiseless observations $y_i = \text{sgn}(\langle x_i, \theta^\star\rangle)$ and ground truth $\theta^\star = (1, 0, \ldots, 0)^\top$. Further, assume that $\lambda > 0$ and $\epsilon = \epsilon_0/\sqrt{d}$, where $\epsilon_0$ is a numerical constant. Let $(\nu_\perp^\star, \nu_\|^\star, r^\star, \delta^\star, \mu^\star, \tau^\star)$ be the unique solution of*

$$\min_{\substack{\nu_\perp \geq 0, \tau \geq 0, \\ \nu_\|, \delta \geq 0}} \max_{\substack{r \geq 0, \\ \mu \geq 0}} \mathbb{E}_{Z_\|, Z_\perp}\left[\mathcal{M}_\ell(|Z_\|\nu_\| + Z_\perp\nu_\perp - \epsilon_0\delta, \frac{\tau}{r})\right] - \delta\mu + \frac{r\tau}{2} + \lambda(\nu_\perp^2 + \nu_\|^2)$$

$$-\nu_\perp\sqrt{\left[(\mu^2 + \gamma r^2) - (\mu^2 + \gamma r^2)\text{erf}(\mu/(\sqrt{\gamma}r\sqrt{2})) - \sqrt{\frac{2}{\pi}}\sqrt{\gamma}r\mu\exp(-\mu^2/(\gamma r^2 2))\right]}. \tag{10}$$

*Then, for $\lambda > 0$ the estimator $\hat{\theta}_\lambda$ from Equation (3) with logistic loss and consistent $\ell_\infty$ attacks satisfies asymptotically as $d, n \to \infty$, $d/n \to \gamma$*

$$\frac{1}{\sqrt{d}}\|\Pi_\perp\hat{\theta}_\lambda\|_1 \to \delta^\star \quad \text{and} \quad \langle\hat{\theta}_0(\epsilon), \theta^\star\rangle \to \nu_\|^\star \quad \text{and} \quad \|\hat{\theta}_\lambda\|_2^2 \to \nu_\|^{\star 2} + \nu_\perp^{\star 2}. \tag{11}$$

*The convergences hold true in probability.*

For $\lambda > 0$, the Equation (9) has a unique minimizer. In contrast, for $\lambda = 0$ the minimizer of Equation (9) is not unique since the data is robustly seperable. As discussed in Section 3, for $\lambda = 0$ we study instead the penalized max-margin solution from Equation (2). The asymptotic behavior of the corresponding solution is characterized in Theorem D.2 with proof in Appendix D.4.

**Theorem D.2.** *Assume that the we have i.i.d. random features $x_i$ drawn from an isotropic Gaussian with noiseless observations $y_i = \text{sgn}(\langle x_i, \theta^\star\rangle)$ and $\theta^\star = (1, 0, \cdots, 0)^\top$ as described in Section 3. Further, assume that $\epsilon = \epsilon_0/\sqrt{d}$. Let $(\nu_\perp^\star, \nu_\|^\star, r^\star, \delta^\star, \zeta^\star, \kappa^\star)$ be the unique solution of*

$$\min_{\substack{\nu_\perp \geq 0, \\ \delta \geq 0, \kappa, \nu_\|}} \max_{\zeta, r \geq 0} \nu_\|^2 - \kappa\nu_\perp^2 - \delta\zeta - \frac{\gamma r^2}{4(1 + \kappa)} + r\sqrt{\mathbb{E}_{Z_\|, Z_\perp}\left[\max\left(0, 1 + \epsilon_0\delta - |Z_\|\nu_\| + Z_\perp\nu_\perp|\right)^2\right]}$$

$$+ \frac{1}{2(1 + \kappa)}\left(\frac{\gamma r^2 + \zeta^2}{2}\text{erf}\left(\frac{\zeta}{\sqrt{2}\sqrt{\gamma}r}\right) - \frac{\zeta^2}{2} + \frac{\sqrt{\gamma}r\zeta}{\sqrt{2\pi}}\exp\left(-\frac{\zeta^2}{2\gamma r^2}\right)\right). \tag{12}$$

*Then, the estimator $\hat{\theta}_0(\epsilon)$ from Equation* (2) *with logistic loss and consistent $\ell_\infty$ attacks satisfies asymptotically as $d, n \to \infty$, $d/n \to \gamma$*

$$\frac{1}{\sqrt{d}}\|\Pi_\perp \hat{\theta}_0(\epsilon)\|_1 \to \delta^\star \text{ and } \langle \hat{\theta}_0(\epsilon), \theta^\star \rangle \to \nu_\|^\star \text{ and } \|\hat{\theta}_0(\epsilon)\|_2^2 \to \nu_\|^{\star 2} + \nu_\perp^{\star 2}. \tag{13}$$

*The convergences hold true in probability.*

**Remark D.3.** *Theorem* 3.1 *is obtained from Theorem* D.1,D.2 *when inserting the expression from Equation* (11),(13) *into the expression of the risk in Lemma* A.1.

### D.1. Solving the optimization problems

To obtain concrete predictions from Theorems D.1 and D.2 we need to solve the corresponding optimization problems. Using the fact that the objective is concave in $r, \mu$ respectively $r, \zeta$ and convex in all other variables, we find the unique solution by solving the systems of equations stated in Corollary D.4,D.5 presented below — assuming that the solution is not obtained on the boundary. For numerical predictions, we solve the systems of equations using a least squares solver in *MATLAB* where we approximated the expectations using a standard numerical integrator.

**Regularized estimate ($\lambda > 0$):** We make use of the following well known facts on the derivative of the Moreau envelope (e.g. see (Jourani et al., 2014)):

$$\frac{\partial \mathcal{M}_f(x, t)}{\partial x} = \frac{1}{t}(x - \operatorname{Prox}_f(x, t)) \text{ and } \frac{\partial \mathcal{M}_f(x, t)}{\partial t} = -\frac{1}{2t^2}(x - \operatorname{Prox}_f(x, t))^2$$

where $\operatorname{Prox}_f(x, \mu)$ is the proximal operator

$$\operatorname{Prox}_f(x, \mu) = \arg\min_t f(t) + \frac{1}{2\mu}(t - x)^2.$$

Let $M = \mathbb{E}_{Z_\|, Z_\perp}\left[\mathcal{M}_\ell(Z_\| \nu_\| + Z_\perp \nu_\perp - \epsilon\delta, \frac{\tau}{r})\right]$ and denote with

$$M_{\nu_\|} = \mathbb{E}_{Z_\|, z_\perp}\left[\frac{\partial_x \mathcal{M}_\ell}{\partial x}(Z_\| \nu_\| + Z_\perp \nu_\perp - \epsilon\delta, \frac{\tau}{r})\frac{\partial(Z_\| \nu_\| + Z_\perp \nu_\perp - \epsilon\delta)}{\partial \nu_\|}\right]$$

the derivative of the expected Moreau envelope with respect to $\nu_\|$ and similar for the other variables.

Further, define

$$g := (\mu^2 + \gamma r^2) - (\mu^2 + \gamma r^2)\operatorname{erf}(\mu/(\sqrt{\gamma}r\sqrt{2})) - \sqrt{\frac{2}{\pi}}\sqrt{\gamma}r\mu \exp(-\mu^2/(\gamma r^2 2))$$

and denote with $g_\mu$ respectively $g_r$ the corresponding partial derivatives. This allows us to express the optimization objective from Theorem D.1 as

$$C = M - \nu_\perp \sqrt{g} - \delta\mu + \frac{r\tau}{2} + \lambda(\nu_\perp^2 + \nu_\|^2).$$

Since the problem is concave in $r, \mu$ and convex in the other variables, we can equivalently solve the system of equations that results from setting the first order conditions, i.e. $\nabla C = 0$, which gives us the following corollary.

**Corollary D.4.** *Assume that the solution of Equation* (10) *from Theorem* D.1 *is not obtained on the boundary. Then, the*

*solution satisfies the following system of Equations.*

$$0 = \frac{\partial}{\partial \nu_\|} C = M_{\nu_\|} + 2\lambda\nu_\|$$

$$0 = \frac{\partial}{\partial \nu_\perp} C = M_{\nu_\perp} + 2\lambda\nu_\perp - \sqrt{g}$$

$$0 = \frac{\partial}{\partial r} C = M_r - \frac{\nu_\perp}{2\sqrt{g}} g_r + \frac{\tau}{2}$$

$$0 = \frac{\partial}{\partial \delta} C = M_\delta - \mu$$

$$0 = \frac{\partial}{\partial \mu} C = -\frac{\nu_\perp}{2\sqrt{g}} g_\mu - \delta$$

$$0 = \frac{\partial}{\partial \tau} C = M_\tau + \frac{r}{2}$$

**Robust max-$\ell_2$-margin interpolator ($\lambda = 0$):** Similarly, we can also reformulate the optimization problem in Theorem D.2. For brevity of notation, let $T = \mathbb{E}_{Z_\|, Z_\perp} \left[ \max\left(0, 1 + \epsilon_0\delta - Z_\|\nu_\| + Z_\perp\nu_\perp\right)^2 \right]$ and

$$H = \left( \frac{\gamma r^2 + \zeta^2}{2} \operatorname{erf}\left(\frac{\zeta}{\sqrt{2}\sqrt{\gamma}r}\right) - \frac{\zeta^2}{2} + \frac{\sqrt{\gamma}r\zeta}{\sqrt{2\pi}} \exp\left(-\frac{\zeta^2}{2\gamma r^2}\right) \right).$$

Further, denote with $T_{\nu_\perp}$ the derivative of $T$ with respect to $\nu_\perp$ and similarly for the other variables. We use the same notation for the partial derivatives of $H$. Finally, we can write the optimization objective from Theorem D.2 as

$$C = \nu_\|^2 - \kappa\nu_\perp^2 - \delta\zeta - \frac{\gamma r^2}{4s} + 2sH + r\sqrt{T} + \frac{\tau_r \kappa}{2}.$$

Since, once again, the solution of the optimization problem must satisfy $\nabla C = 0$ if it does not lie on the boundary, we analogously obtain the following corollary.

**Corollary D.5.** *Assume that the solution of Equation (12) from Theorem D.2 is not obtained on the boundary. Then, the solution satisfies the following system of Equations.*

$$0 = \frac{\partial}{\partial \nu_\|} C = 2\nu_\| + \frac{1}{2\sqrt{T}} r T_{\nu_\|}$$

$$0 = \frac{\partial}{\partial \nu_\perp} C = -2\kappa\nu_\perp + \frac{1}{2\sqrt{T}} r T_{\nu_\perp}$$

$$0 = \frac{\partial}{\partial r} C = \sqrt{T} + H_r - \frac{\gamma r}{2(1 + \kappa)}$$

$$0 = \frac{\partial}{\partial \delta} C = -\zeta + \frac{1}{2\sqrt{T}} r T_\delta$$

$$0 = \frac{\partial}{\partial \zeta} C = -\delta + H_y$$

$$0 = \frac{\partial}{\partial \kappa} C = -\nu_\perp^2 + H_\kappa + \frac{\gamma r^2}{4(1 + \kappa)^2}$$

### D.2. Label noise

While our results assume noiseless observations $y_i = \operatorname{sgn}(\langle x_i, \theta^\star \rangle)$, Theorem D.1,D.2 can be extended to the case where additional label noise is added to the observations. That is, it can be extended to the case where $y_i = \operatorname{sgn}(\langle x_i, \theta^\star \rangle)\xi_i$ with $\xi_i$ i.i.d. distributed and $\mathbb{P}(\xi_i = 1) = 1 - \sigma$ $\mathbb{P}(\xi_i = -1) = \sigma$ with label noise $\sigma$.

Note that as discussed in Section 4.2, the robust max-margin solution (2) might not exists for noisy observations. In this case, the estimate (3) has a unique solution also for $\lambda = 0$. In fact, asymptotically we can find a threshold $\gamma^\star$ such that for any $\gamma < \gamma^\star$ the robust max-margin solution does not exists and for any $\gamma \geq \gamma^\star$ the robust max-margin solution exists. The threshold can be found using the CGMT when following the same argument as in Theorem 6.1 (Javanmard & Soltanolkotabi, 2020).

Finally, we remark that when $\lambda > 0$ or $\lambda = 0$ and $\gamma < \gamma^\star$ we can apply Theorem D.1 when replacing $|Z_\|\|$ with $\xi|Z_\|\|$. Similarly, for $\lambda = 0$ and $\gamma \geq \gamma^\star$ we can apply Theorem D.2 when replacing $|Z_\|\|$ with $\xi|Z_\|\|$ where $\xi$ is drawn form the same distribution as $\xi_i$ defined above.

### D.3. Proof of Theorem D.1

The proof is similar to the proof of Theorem 6.4 (Javanmard & Soltanolkotabi, 2020). Denote with $X \in \mathbb{R}^{n \times d}$ the input data matrix and with $y \in \mathbb{R}^n$ the vector containing the observations. Recall that the estimator $\hat{\theta}$ is given by

$$\hat{\theta} = \arg\min_\theta \frac{1}{n} \sum_{i=1}^n \ell(y_i \langle x_i, \theta \rangle - \epsilon \|\Pi_\perp \theta\|_1) + \lambda \|\theta\|_2^2$$

$$= \arg\min_\theta \frac{1}{n} \sum_{i=1}^n \ell(v_i - \epsilon \|\Pi_\perp \theta\|_1) + \lambda \|\theta\|_2^2 \text{ such that } v = D_y X \theta, \tag{14}$$

where $\ell(x) = \log(1 + \exp(-x))$ is the logistic loss, $X \in \mathbb{R}^{n \times d}$ is the data matrix and $D_y$ the diagonal matrix with entries $(D_y)_{i,i} = y_i$. We can then introduce the Lagrange multipliers $u \in \mathbb{R}^n$ to obtain:

$$\min_{\theta,v} \max_u \frac{1}{n} \sum_{i=1}^n \ell(v_i - \epsilon \|\Pi_\perp \theta\|_1) + \frac{1}{n} u^\top D_y X \theta - \frac{1}{n} u^\top v + \lambda \|\theta\|_2^2.$$

Furthermore, we can separate $X = X\Pi_\perp + X\Pi_\|$, which gives us:

$$\min_{\theta,v} \max_u \frac{1}{n} \sum_{i=1}^n \ell(v_i - \epsilon \|\Pi_\perp \theta\|_1) + \frac{1}{n} u^\top D_y X\Pi_\| \theta + \frac{1}{n} u^\top D_y X\Pi_\perp \theta - \frac{1}{n} u^\top v + \lambda \|\theta\|_2^2. \tag{15}$$

**Convex Gaussian Minimax Theorem**    We can now make use of the CGMT, which states that

$$\min_{\theta \in U_\theta} \max_{u \in U_u} u^\top X \theta + \psi(u, \theta), \tag{16}$$

with $\psi$ convex in $\theta$ and concave in $u$ has asymptotically pointwise the same solution when $d, n \to \infty$, $d/n \to \gamma$ as

$$\min_{\theta \in U_\theta} \max_{u \in U_u} \|u\|_2 g^\top \theta + u^\top h \|\theta\|_2 + \psi(u, \theta), \tag{17}$$

with $g \in \mathbb{R}^d$, $h \in \mathbb{R}^n$ random vectors with i.i.d. standard normal distributed entries and compact sets $U_\theta$ and $U_u$. As commonly referred to in the literature, we call Equation (24) the primal optimization problem and Equation (17) the auxiliary optimization problem. We omit the precise statement and refer the reader to (Thrampoulidis et al., 2015). The CGMT has already been used in several works studying high dimensional asymptotic logistic regression (Salehi et al., 2019) and also when training with adversarial attacks (Javanmard & Soltanolkotabi, 2020). Essentially, the application of the CGMT is based on the following facts:

1. The objective (15) is concave in $u$ and convex in $v, \theta$.

2. We can restrict $u, v, \theta$ to compact sets without changing the solution. For $\theta$ we note that this is a consquence of $\lambda > 0$ and for $u$ we note that the stationary condition requires $u_i = \ell'(v_i - \epsilon \|\Pi_\perp \theta\|_1)$.

3. $X\Pi_\perp$ is independent of the observations $y$ and of $X\Pi_\|$.

As a result, the CGMT states that the solution of the primal optimizatin problem (15) asymptotically concentrates around the same value the solution of the following auxiliary optimization problem

$$\min_{\theta,v} \max_{u} \frac{1}{n} \sum_{i=1}^{n} \ell(v_i - \epsilon \|\Pi_\perp \theta\|_1) + \frac{1}{n} u^\top D_y X \Pi_\| \theta + \frac{1}{n} \|u^\top D_y\|_2 g^\top \Pi_\perp \theta$$
$$+ \frac{1}{n} u^\top D_y h \|\Pi_\perp \theta\|_2 - \frac{1}{n} u^\top v + \lambda \|\theta\|_2^2,$$

where $g \in \mathbb{R}^d$ and $h \in \mathbb{R}^n$ are vectors with i.i.d. standard normal distributed entries.

**Scalarization of the optimization problem** We now aim to simplify the optimization problem. In a first step, we maximize over $u$. For this, define $r = \|u\|_2/\sqrt{n}$, which allows us to equivalently write:

$$\min_{\theta,v} \max_{r \geq 0} \frac{1}{n} \sum_{i=1}^{n} \ell(v_i - \epsilon \|\Pi_\perp \theta\|_1) + \frac{r}{\sqrt{n}} \|D_y X \Pi_\| \theta + D_y h \|\Pi_\perp \theta\|_2 - v\|_2 + \frac{1}{\sqrt{n}} r g^\top \Pi_\perp \theta + \lambda \|\theta\|_2^2,$$

where we have used the fact that $\|u^\top D_y\|_2 = \|u\|_2$. In order to proceed, we want to separate $\Pi_\perp \theta$ form the loss $\ell(v, \Pi_\perp \theta) := \frac{1}{n} \sum_{i=1}^{n} \ell(v_i - \epsilon \|\Pi_\perp \theta\|_1)$. Denoting by $\tilde{\ell}$ the conjugate of $\ell$ we can write $\ell(v, \Pi_\perp \theta)$ in terms of its conjugate with respect to $\Pi_\perp \theta$:

$$\ell(v, \Pi_\perp \theta) = \sup_{w} \frac{1}{\sqrt{d}} w^\top \Pi_\perp \theta - \tilde{\ell}(v, w)$$
$$= \sup_{w} \frac{1}{\sqrt{d}} w^\top \Pi_\perp \theta - \sup_{\delta \geq 0} \left( \frac{\sqrt{d}}{\sqrt{d}} \delta \|w\|_\infty - \frac{1}{n} \sum_{i=1}^{n} \ell(v_i - \sqrt{d}\epsilon \delta) \right)$$
$$= \sup_{w} \inf_{\delta \geq 0} \frac{1}{\sqrt{d}} w^\top \Pi_\perp \theta - \delta \|w\|_\infty + \frac{1}{n} \sum_{i=1}^{n} \ell(v_i - \epsilon_0 \delta),$$

where for the second identity we use the derivation for the conjugate of $\ell$ from Lemma A.2 in the paper (Javanmard & Soltanolkotabi, 2020).

Next, we note that because the problem is convex in $r$ and concave in $\theta, v$, we can swap maximization with minimization. Interchanging the order of maximization over $w$ and minimization over $\delta$, we get

$$\max_{r \geq 0} \min_{\substack{\theta,v, \\ \delta \geq 0}} \frac{1}{n} \sum_{i=1}^{n} \ell(v_i - \epsilon_0 \delta) + \frac{r}{\sqrt{n}} \left\| D_y X \Pi_\| \theta + D_y h \|\Pi_\perp \theta\|_2 - v \right\|_2 + \lambda \|\theta\|_2^2 \tag{18}$$
$$+ \max_{w} \left[ \frac{1}{\sqrt{d}} w^\top \Pi_\perp \theta - \delta \|w\|_\infty + \frac{1}{\sqrt{n}} r g^\top \Pi_\perp \theta \right]$$

where we used again the fact that we can interchange the order maximization and minimization. Next, we simplify the optimization over $\theta$. Write $\Pi_\| \theta = \Pi_\| 1 \nu_\|$ where $\nu_\| \in \mathbb{R}$ and let $\nu_\perp = \|\Pi_\perp \theta\|_2$. We can simplify

$$\max_{r \geq 0} \min_{\substack{\nu_\perp \geq 0, \\ \delta \geq 0, \\ \nu_\|,v}} \frac{1}{n} \sum_{i=1}^{n} \ell(v_i - \epsilon_0 \delta) + \frac{r}{\sqrt{n}} \|D_y X \Pi_\| 1 \nu_\| + D_y h \nu_\perp - v\|_2 + \lambda(\nu_\|^2 + \nu_\perp^2) \tag{19}$$
$$+ \max_{w} \left[ -\frac{1}{\sqrt{d}} \nu_\perp \|\Pi_\perp (w - \sqrt{\gamma} r g)\|_2 - \delta \|w\|_\infty \right]$$

In order to obtain a low dimensional scalar optimization problem, we still need to scalarize the optimization over $w$ and $v$. For this, we replace the term $\|D_y X \Pi_\| 1 \nu_\| + D_y h \nu_\perp - v\|_2$ with its square, which is achieved by using the following

identity $\min_{\tau \geq 0} \frac{x^2}{2\tau} + \frac{\tau}{2} = x$. Hence,

$$\max_{\substack{r \geq 0}} \min_{\substack{\nu_\perp \geq 0, \tau \geq 0 \\ \delta \geq 0, \\ \nu_\parallel, v}} \frac{1}{n} \sum_{i=1}^n \ell(v_i - \epsilon_0 \delta) + \frac{r}{2\tau n} \|D_y X \Pi_\parallel 1 \nu_\parallel + D_y h \nu_\perp - v\|_2^2 + \frac{\tau r}{2} + \lambda(\nu_\parallel^2 + \nu_\perp^2)$$

$$+ \max_w \left[ -\frac{1}{\sqrt{d}} \nu_\perp \|\Pi_\perp (w - \sqrt{\gamma} r g)\|_2 - \delta \|w\|_\infty \right]$$

interchanging again the order of maximization and minimization, we can now separately solve the following two inner optimization problems:

$$\max_w \quad -\nu_\perp \frac{1}{d} \|\Pi_\perp (w - \sqrt{\gamma} r g)\|_2 - \delta \|w\|_\infty \tag{20}$$

$$\min_v \quad \frac{r}{2\tau n} \|D_y X \Pi_\parallel 1 \nu_\parallel + D_y h \nu_\perp - v\|_2^2 + \sum_{i=1}^n \ell(v_i - \epsilon_0 \delta) \tag{21}$$

**Equation** (20)  Let $\mathrm{ST}_t(x) = \begin{cases} 0 & |x| \leq t \\ \mathrm{sgn}(x)(|x| - t) & \text{else} \end{cases}$ be the soft threshold function. We have

$$\max_w \quad -\nu_\perp \frac{1}{\sqrt{d}} \|\Pi_\perp (w - \sqrt{\gamma} r g)\|_2 - \delta \|w\|_\infty$$

$$= -\min_w \nu_\perp \frac{1}{\sqrt{d}} \|\Pi_\perp (w - \sqrt{\gamma} r g)\|_2 - \delta \|w\|_\infty$$

$$\overset{\mu = \|w\|_\infty}{=} -\min_{\mu \geq 0} \nu_\perp \sqrt{\frac{1}{d} \sum_{i=2}^d \left( \mathrm{ST}_\mu \left( \sqrt{\gamma} r g_i \right) \right)^2} + \delta \mu$$

$$\overset{\text{LLN as } d \to \infty}{\to} -\min_{\mu \geq 0} \nu_\perp \sqrt{\mathbb{E}_Z \left( \mathrm{ST}_\mu \left( \sqrt{\gamma} r Z \right) \right)^2} + \delta \mu,$$

where we used in the third line that the ground truth $\theta^\star$ is 1-sparse and in the last line that the expectation exists for $Z \sim \mathcal{N}(0, 1)$. Finally, we can further simplify

$$\mathbb{E}_Z \left( \mathrm{ST}_\mu \left( \sqrt{\gamma} r Z \right) \right)^2 = \gamma r^2 \mathbb{E}_Z \left( \mathrm{ST}_{\mu/(\sqrt{\gamma} r)} (Z) \right)^2$$

$$= \gamma r^2 \mathbb{E}_Z (Z - \mu/(\sqrt{\gamma} r))^2 - \mathbb{E}_Z \mathbb{1}_{|Z| \leq \mu/(\sqrt{\gamma} r)} (Z - \mu/(\sqrt{\gamma} r))^2$$

$$= (\mu^2 + \gamma r^2) \left( 1 - \mathrm{erf}(\mu/(\sqrt{2\gamma} r)) \right) - \sqrt{\gamma} r \mu \sqrt{\frac{2}{\pi}} \exp(-\mu/(2\gamma r^2)).$$

Hence, we can conclude the first term.

**Equation** (21)  For the second term we also aim to apply the law of large numbers. We have

$$\min_v \quad \frac{r}{2\tau n} \|D_y X \Pi_\parallel 1 \nu_\parallel + D_y h \nu_\perp - v\|_2^2 + \frac{1}{n} \sum_{i=1}^n \ell(v_i - \epsilon_0 \delta)$$

$$\overset{\tilde{v} = v - \epsilon \delta}{=} \min_{\tilde{v}} \quad \frac{r}{2\tau n} \|D_y X \Pi_\parallel 1 \nu_\parallel + D_y h \nu_\perp - \tilde{v} - \epsilon_0 \delta\|_2^2 + \frac{1}{n} \sum_{i=1}^n \ell(\tilde{v}_i)$$

$$\overset{\tilde{v} = v - \epsilon \delta}{=} \min_{\tilde{v}} \quad \frac{1}{n} \sum_{i=1}^n \frac{r}{2\tau n} \left( (D_y X \Pi_\parallel 1 \nu_\parallel)_i + (D_y h \nu_\perp)_i - \tilde{v}_i - \epsilon_0 \delta \right)^2 + \ell(\tilde{v}_i)$$

$$\overset{\text{LLN}}{\to} \mathbb{E}_{Z_\parallel, Z_\perp} \left[ \mathcal{M}_\ell(|Z_\parallel| \nu_\parallel + Z_\perp \nu_\perp - \epsilon_0 \delta, \frac{\tau}{r}) \right],$$

where in the last line we used that $(D_y X \Pi_\parallel \Pi_\perp 1)_i = y_i x_i^\top \theta^\star = \xi_\sigma \mathrm{sgn}(x_i^\top \theta^\star) x_i^\top \theta^\star$ has the same distribution as $|Z_\parallel|$ with $Z_\parallel \sim \mathcal{N}(0, 1)$. Further, to apply the law of large numbers, we need to show that the Moreau envelope exists. Similarly to Theorem 1 (Salehi et al., 2019), this follows immediately when noting that $\mathcal{M}_\ell(x, \mu) \leq \ell(x) = \log(1 + \exp(-x)) \leq \log(2) + |x|$. Finally, we obtain the desired optimization problem in Equation (10) when combining these results.

**Convergences** Finally, note that the optimum $\delta^\star$ in Equation (18) satisfies $\delta^\star = \frac{1}{\sqrt{d}}\|\Pi_\perp\theta\|$, and similarly the optima $\nu_\perp^\star$ and $\nu_\parallel^\star$ in Equation (19) satisfy $\nu_\perp^\star = \|\Pi_\perp\theta\|_2$ and $\nu_\parallel^\star = \langle\theta,\theta^\star\rangle$. Hence we can conclude the proof as the solutions (10), (14) concentrate asymptotically as $d, n \to \infty$ around the same optima.

## D.4. Proof of Theorem D.2

The proof is similar to the proof of Theorem 6.3 (Javanmard & Soltanolkotabi, 2020). Recall the robust max-margin solution from Equation (2):

$$\min_\theta \|\theta\|_2^2 \text{ such that } \langle\theta, x_i\rangle - \epsilon\|\Pi_\perp\theta\|_1 \geq 1 \text{ for all } i \tag{22}$$

After introducing the Lagrange multipliers $\zeta$ and $u$ we can write equivalently:

$$\min_{\theta,\delta} \max_{\substack{u:u_i\geq 0,\\ \zeta\geq 0}} \|\theta\|_2^2 + \frac{1}{n}u^\top\left(1 + 1\epsilon_0\delta - D_y\mathrm{X}\theta\right) + \zeta\left(\frac{\|\Pi_\perp\theta\|_1}{\sqrt{d}} - \delta\right). \tag{23}$$

and separating again $\mathrm{X} = \mathrm{X}\Pi_\perp + \mathrm{X}\Pi_\parallel$, we get

$$\min_{\theta,\delta} \max_{\substack{u:u_i\geq 0,\\ \zeta\geq 0}} \|\theta\|_2^2 + \frac{1}{n}u^\top\left(1 + 1\epsilon_0\delta - D_y\mathrm{X}\Pi_\parallel\theta - D_y\mathrm{X}\Pi_\perp\theta\right) + \zeta\left(\frac{\|\Pi_\perp\theta\|_1}{\sqrt{d}} - \delta\right) \tag{24}$$

**Convex Gaussian Minimax Theorem** Since the adversarial attacks are consistent and the observations are noiseless, we know the solution in Equation (22) exists for all $d, n$. Yet, in order to apply the CGMT, we have to show that we can restrict $u$ and $\theta$ to compact sets. This follows from a simple trick as explained in Section D.3.1 in (Javanmard & Soltanolkotabi, 2020). Hence, the primal optimization problem from Equation (24) can be asymptotically replaced with the following auxiliary optimization:

$$\min_{\theta,\delta} \max_{\substack{u:u_i\geq 0,\\ \zeta\geq 0}} \|\theta\|_2^2 + \frac{1}{n}u^\top\left(1 + 1\epsilon_0\delta - D_y\mathrm{X}\Pi_\parallel\theta + D_yh\|\Pi_\perp\theta\|_2\right) + \frac{1}{n}\|u\|_2 g^\top\Pi_\perp\theta + \zeta\left(\frac{\|\Pi_\perp\theta\|_1}{\sqrt{d}} - \delta\right) \tag{25}$$

**Scalarization of the optimization problem** The goal is again to scalarize the optimization problem. As a first step, we can solve the optimization over $u$ when defining $r = \frac{\|u\|_2}{\sqrt{n}}$:

$$\min_{\theta,\delta} \max_{\substack{r\geq 0,\\ \zeta\geq 0}} \|\theta\|_2^2 + \frac{r}{\sqrt{n}}\|\max\left(0, 1 + 1\epsilon_0\delta - D_y\mathrm{X}\Pi_\parallel\theta + D_yh\|\Pi_\perp\theta\|_2\right)\|_2$$
$$+ \frac{r\sqrt{\gamma}}{\sqrt{d}}\|u\|_2 g^\top\Pi_\perp\theta + \zeta\left(\frac{1}{\sqrt{d}}\|\Pi_\perp\theta\|_1 - \delta\right),$$

where $\max$ is taken elementwise over the vector. We now want to separate $\|\Pi_\perp\theta\|_2$ from the term in $\max(.)$. This is achieved by introducing the variable $\nu_\perp \geq 0$ and the Lagrange multiplier $\kappa$. Further, we set $\nu_\parallel = \langle\theta^\star, \Pi_\parallel\theta\rangle$, which allows us to equivalently write

$$\min_{\substack{\nu_\perp\geq 0,\\ \nu_\parallel,\delta}} \max_{\substack{r\geq 0,\\ \zeta\geq 0,\kappa}} \nu_\parallel^2 + \|\Pi_\perp\theta\|_2^2 + \kappa(\|\Pi_\perp\theta\|_2^2 - \nu_\perp^2) + \frac{r}{\sqrt{n}}\|\max\left(0, 1 + 1\epsilon_0\delta - D_y\mathrm{X}\Pi_\parallel\theta_0\nu_\parallel + D_yh\nu_\perp\right)\|_2$$
$$+ \frac{r\sqrt{\gamma}}{\sqrt{d}}\|u\|_2 g^\top\Pi_\perp\theta + \zeta\left(\frac{1}{\sqrt{d}}\|\Pi_\perp\theta\|_1 - \delta\right) \tag{26}$$

We further note that

$$\frac{r}{\sqrt{n}}\|\max\left(0, 1 + 1\epsilon_0\delta - D_y\mathrm{X}\Pi_\parallel\theta_0\nu_\parallel + D_yh\nu_\perp\right)\|_2$$
$$\overset{\mathrm{LLN}}{\to} r\sqrt{\mathbb{E}_{Z_\parallel,Z_\perp}\left[\max\left(0, 1 + \epsilon_0\delta - |Z_\parallel|\nu_\parallel + Z_\perp\nu_\perp\right)^2\right]} =: \sqrt{T}.$$

Next, by completion of the squares we get

$$\min_{\substack{\nu_\perp \geq 0, \ r \geq 0, \\ \nu_\parallel, \delta \ \ \zeta \geq 0, \kappa}} \max \nu_\parallel^2 + (1+\kappa)\|\Pi_\perp \theta + \frac{r\sqrt{\gamma}}{\sqrt{d}2(1+\kappa)}g\|_2^2 - \frac{r^2\gamma}{4(1+\kappa)}\|g/\sqrt{d}\|_2^2$$

$$- \kappa\nu_\perp^2 + \sqrt{T} + \zeta\left(\frac{1}{\sqrt{d}}\|\Pi_\perp \theta\|_1 - \delta\right),$$

with $\|g/\sqrt{d}\|_2^2 \to 1$. Hence, it only remains to solve the inner optimization over $\Pi_\perp \theta$. We can write:

$$\min_{\Pi_\perp \theta} \ (1+\kappa)\|\Pi_\perp \theta + \frac{r\sqrt{\gamma}}{\sqrt{d}2(1+\kappa)}g\|_2^2 + \zeta\frac{\|\Pi_\perp \theta\|_1}{\sqrt{d}}$$

$$\overset{\tilde{\theta}_\perp = \frac{\Pi_\perp \theta}{\sqrt{d}}}{=} \min_{\tilde{\theta}_\perp} \frac{1}{d}(1+\kappa)\|\tilde{\theta}_\perp + \frac{r\sqrt{\gamma}}{2(1+\kappa)}g\| + \zeta\frac{\|\Pi_\perp \theta\|_1}{d}$$

$$= \frac{1}{d}\sum_{i=2}^{d} \min_{(\tilde{\theta}_\perp)_i} \ (1+\kappa)((\tilde{\theta}_\perp)_i + \frac{r\sqrt{\gamma}}{2(1+\kappa)}g_i)^2 + \zeta|(\tilde{\theta}_\perp)_i|$$

$$= \frac{1}{d}2(1+\kappa)\sum_{i=2}^{d} \min_{(\tilde{\theta}_\perp)_i} \ \frac{1}{2}((\tilde{\theta}_\perp)_i + \frac{r\sqrt{\gamma}}{2(1+\kappa)}g_i)^2 + \frac{\zeta}{2(1+\kappa)}|(\tilde{\theta}_\perp)_i|$$

$$= \frac{1}{d}2(1+\kappa)\sum_{i=2}^{d} \ell_H(-\frac{r\sqrt{\gamma}}{2(1+\kappa)}g_i, \frac{\zeta}{2(1+\kappa)})$$

$$\to 2(1+\kappa)\mathbb{E}_Z \, \ell_H\left(\frac{r\sqrt{\gamma}}{2(1+\kappa)}Z, \frac{\zeta}{2(1+\kappa)}\right)$$

where $\ell_H$ is the Huber loss, given by $\ell_H(x,y) = \begin{cases} 0.5x^2 & |x| \leq y \\ y(|x| - 0.5y) \end{cases}$. Finally, we can conclude the proof from

$$\mathbb{E}_Z \, \ell_H(aZ, b) = \frac{a^2 + b^2}{2}\mathrm{erf}\left(\frac{b}{\sqrt{2}a}\right) - \frac{b^2}{2} + \frac{ab}{\sqrt{2\pi}}\exp\left(-\frac{b^2}{2a^2}\right)$$

**Convergence** We note again that the optimum $\delta^\star$ in Equation (23) satisfies $\delta^\star = \frac{1}{\sqrt{d}}\|\Pi_\perp \theta\|$, and similarly the optima $\nu_\perp^\star$ and $\nu_\parallel^\star$ in Equation (26) satisfy $\nu_\perp^\star = \|\Pi_\perp \theta\|_2$ and $\nu_\parallel^\star = \langle\theta, \theta^\star\rangle$. Hence we can conclude the proof as the solutions (24), (25) concentrate asymptotically as $d, n \to \infty$ around the same optima.