# OpenReview forum: "Maximizing the robust margin provably overfits on noiseless data"
_ICML.cc/2021/Workshop/AML — ICML 2021 Workshop AML Poster_

### Official Review · Reviewer_4T65 · 2021-06-20
**A good paper providing theoretical and empirical evidence for understanding robust over-fitting**

**Rating:** Accept
**Confidence:** 3

**Review:**

The paper shows that robust overfitting can not be attributed to noise or problematic samples in the training data set, and actually, it may arise from the lack of regularization. The authors prove that ridge regularization of the robust logistic loss leads to a solution that generalizes better than the unregularized robust max-margin estimator obtained by gradient descent. And empirically, early stopping gradient descent yields a similar benefit. The authors suggest introducing wrong labels in the training loss as it prevents the estimator from converging to the max-margin solution.

Basically, the paper is well-motivated and theoretically sound, providing scientific insights for various communities. I am looking forward to seeing a complete version of it in the near future.

---

### Decision · Program_Chairs · 2021-06-21

**Decision:**

Accept (Poster)

**Comment:**

A good paper showing that robust overfitting can not be attributed to noise or problematic samples in the training data set. The paper is well-motivated and theoretically sound, providing scientific insights for various communities.